# Segmenting Increasing- and High-Risk Alcohol Drinkers by Motives and Occasions: Implications for Targeted Interventions

**DOI:** 10.3390/nu17233745

**Published:** 2025-11-28

**Authors:** Chloe Bennett, Liam J. Barratt, Alastair O’Brien

**Affiliations:** 1Division of Medicine, University College London, London WC1E 6BT, UK; chloe.bennett.24@ucl.ac.uk; 2Drinkaware, London EC1Y 4SE, UK; lbarratt@drinkaware.co.uk

**Keywords:** alcohol misuse, public health, drinking motives, drinking occasions, cluster analysis, symptoms of dependency, loneliness

## Abstract

Background/Objectives: Excess alcohol consumption remains a major challenge for public health in the UK. This study aimed to define and characterise clusters of increasing- and high-risk drinkers, based on drinking motives and occasions and contextualised by demographics and psychosocial factors, to inform tailored harm-reduction strategies. Methods: Secondary analysis of the nationally representative Drinkaware Monitor 2023 survey (*n* = 10,473) identified six clusters of increasing- and high-risk drinkers (*n* = 2486 weighted). Segmentation was based on drinking occasions and drinking motives using the Drinking Motives Questionnaire for Adults (DMQ-A). Clusters were then characterised using demographic factors, drinking patterns, and psychosocial indicators (dependency symptoms, loneliness, and risk perception), with weighted two-proportion z-tests and false discovery rate correction applied. Results: Six distinct subgroups emerged. Cluster 1 consisted of older drinkers (55+) motivated by product and taste, with fewer dependency symptoms and lower loneliness. Cluster 2 showed lower social motives but greater likelihood of drinking alone at home and with meals. Cluster 3, younger and more diverse, displayed very high social and identity motives (e.g., drinking to feel confident), alongside elevated loneliness and dependency indicators. Cluster 4 also presented strong social motives but had fewer product and taste motives and more adverse outcomes. Cluster 5, typically older and C2DE, reported higher coping motives (e.g., drinking to unwind) and more solitary drinking. Cluster 6 had consistently lower motives, a narrower range of drinking occasions, and a mixed dependency profile. Conclusions: Increasing- and high-risk drinkers are heterogeneous, shaped by different motives and contexts. Tailoring interventions to subgroup profiles, such as socially motivated younger adults and older adults whose drinking is driven by coping motives, may strengthen alcohol harm-reduction strategies.

## 1. Introduction

Alcohol misuse is a leading cause of premature mortality in the UK, and the second largest individual risk factor for disability-adjusted life years among people aged 15–49 [1]. In 2023, 10,473 deaths were attributed to alcohol-specific causes across the UK, the highest figure since records began [2]. Alcohol-related mortality, which includes partially alcohol-attributable conditions, also rose by 8.8% between 2019 and 2022 [3]. Part of this rise may reflect wider behavioural and service-level changes during the COVID-19 pandemic, including reduced access to healthcare and increases in alcohol consumption among some groups. For example, the 2020 Drinkaware Monitor reported that 11% of drinkers increased their consumption during the pandemic period [4]. The latest data indicates that alcohol-related conditions account for over 486,000 potential years of life lost in England, with rates continuing to rise since 2019 [5].

Alcohol contributes causally to over 200 medical conditions, including cancers and mental health disorders [6], with alcoholic liver disease responsible for nearly three-quarters of alcohol-specific deaths in the UK [7]. The effects of alcohol extend to nearly every internal system, including the brain, liver, pancreas, endocrine, immune and circulatory systems [8]. Classified as a Group 1 carcinogen, the World Health Organisation (WHO) has confirmed that no level of alcohol consumption is safe, though risk increases with dose, frequency and individual health status [9].

Patterns of consumption are typically measured using the Alcohol Use Disorders Identification Test (AUDIT), which provides a risk score reflecting both drinking behaviour and the likelihood of experiencing alcohol-related harm [10]. The latest Health Survey for England reported that 20% of 16–24-year-olds and 14% of 45–54-year-olds were classified as increasing-, higher-risk, or possibly dependent drinkers by the AUDIT [11]. Together, these groups represent almost a quarter of the UK population [12], emphasising the scale of potential harm. Despite this, public health interventions have often focused on dependent drinkers through treatment services or whole-population measures such as pricing and availability, with comparatively less emphasis on the large group of increasing- and high-risk drinkers. For example, minimum unit pricing (MUP) sets a legal lower limit on the price of alcohol, ensuring no drink can be sold below a specified unit threshold [13]. MUP was introduced in Scotland in 2018 (50p per unit) and in Wales in 2020, primarily targeting cheap, high-strength alcohol. Evaluations in Scotland show reductions in alcohol sales, particularly among heavier drinkers, alongside a decline in alcohol-specific hospital admissions and deaths [14,15]. Early evidence from Wales indicates similar effects [16]. These examples highlight the potential of whole-population pricing policies but also emphasise the need for complementary approaches tailored to the heterogeneity of increasing- and high-risk drinkers.

Targeted interventions for higher-risk drinkers, such as Identification and Brief Advice (IBA), are recognised as successful and cost-efficient [17]. IBA consists of a brief, structured conversation in which a health professional screens for hazardous drinking and provides tailored feedback and advice. While effective at the individual level, Public Health England (PHE) emphasises that IBA only achieves meaningful population impact when delivered consistently and at scale. In practice, implementation remains limited, with delivery often inconsistent across services, meaning current provision falls short of population needs [18]. At the same time, alcohol-related hospital admissions remain disproportionately concentrated in more deprived socioeconomic groups [11,17]. This pattern cannot be explained simply by higher consumption in these groups; rather, it reflects the alcohol harm paradox, whereby disadvantaged communities experience greater harm at equivalent or even lower levels of consumption compared with more affluent groups [19]. This highlights the need for strategies that go beyond dependence and address the broader spectrum of harmful alcohol use.

The prevention paradox [20] further illustrates that while high-risk or dependent drinkers face the greatest personal danger, most alcohol-related harm arises from the much larger population of moderate and heavy episodic drinkers [21,22]. For example, O’Dwyer et al. discovered that although dependent drinkers in Ireland had the highest individual risk of harm, 59% of all harms occurred among monthly or occasional heavy episodic drinkers [21,22]. The alcohol harm paradox adds a further dimension: harm is not evenly distributed but is disproportionately borne by those in disadvantaged socioeconomic groups. Together, these paradoxes emphasise that effective alcohol policy must go beyond targeting dependent drinkers alone.

Current UK alcohol-harm-reduction strategies include population-level measures (Chief Medical Officer guidelines, awareness campaigns, price and availability restrictions), primary care targeted interventions (screening, followed by brief advice), and specialist treatment services for dependence [23,24,25]. However, increasing- and high-risk drinkers are a heterogeneous group, and broad messaging may fail to resonate with them.

Segmentation can support the development of more tailored and effective strategies. Earlier analyses of Drinkaware Monitor data grouped drinkers by drinking motivations and occasions but relied on the Drinking Motives Questionnaire—Revised Short Form (DMQ-R SF) [26], a tool developed for an adolescent population, to achieve this. This reliance may have overlooked important drivers of adult drinking. The present study is the first to address this gap by combining drinking occasion data with the Drinking Motives Questionnaire for Adults (DMQ-A), an adult-adapted motives questionnaire [27], in a novel approach to identify six distinct clusters of increasing- and high-risk drinkers [28].

The aim of this study was to define and characterise these clusters, with drinking motives and occasions as the primary factors underpinning segmentation, and additional variables such as demographics, drinking patterns, and adverse outcomes included to provide context. Specifically, the objectives were to (1) describe how drinking motives and occasions vary across clusters compared with the benchmark population of increasing- and high-risk drinkers; (2) explore how demographic and behavioural factors contextualise these clusters; and (3) identify distinguishing features that may inform targeted harm-reduction strategies. It was hypothesised that clusters would show clear and systematic differences, indicating that interventions tailored to motives and drinking contexts could be more effective than generic approaches.

## 2. Methods

### 2.1. Study Design

This study undertook a secondary analysis of cross-sectional data from the Drinkaware Monitor 2023, commissioned by Drinkaware and conducted by YouGov [29]. The survey is provided in full in Appendix A.

Participants were recruited via email invitations from YouGov’s established online research panel, which covers a broad range of demographic groups. This allowed YouGov to select a nationally representative sample across all four UK nations. Fieldwork took place in 2023, with responses collected through YouGov’s secure online survey platform, using unique survey links for each participant.

In total, 10,473 individuals completed the survey across England (6948), Wales (1302), Scotland (1565), and Northern Ireland (658). To allow robust analysis, the survey included larger samples from areas of highest deprivation (IMD deciles 1–2; *n* = 2349) and the lowest (IMD deciles 9–10; *n* = 2580). The dataset was weighted to reflect the regional, age, and social grade distributions within each UK country, with an additional weight applied to align with the overall UK population. All demographic information used for weighting derived from UK Census information by the Office for National Statistics and social grade definitions from the National Readership Survey. ABC1 represents higher and intermediate managerial, administrative and professional occupations, and C2DE represents skilled manual, semi-skilled and unskilled occupations, including those unemployed [29].

### 2.2. Participants

The analysis focused on respondents categorised as increasing- or high-risk drinkers according to the AUDIT; AUDIT scores of 0–7 signify low risk, 8–15 increasing risk, 16–19 higher risk, and ≥20 possible dependence [10]. In total, 2464 respondents were classified as increasing- or high-risk drinkers.

### 2.3. Measures

Segmentation was based on two domains of drinking behaviour: (i) drinking occasions and typical locations (A5_new), and (ii) drinking motivations (A11), assessed using the DMQ-A, which covers social, conformity, and coping motives [27]. Hierarchical clustering using Euclidean distance on scaled data identified six distinct clusters of increasing- and high-risk drinkers. Six clusters were produced after inspection of the clustering dendrogram and cutting at a height of 12. This number of clusters was retained as groups were distinguishable from one another, without the number of clusters becoming excessive. Given the intention of this exploratory study was to examine broad clusters of drinkers with a view to developing personas, a higher number of more distinct clusters was opted against for a smaller number of clusters that would be more practically useful. Of course, more nuanced clusters could be generated by cutting the dendrogram at a lower height—something which could be explored in future work.

Once defined, clusters were profiled against additional variables from the Drinkaware Monitor, including demographics, drinking patterns, symptoms of dependence, adverse outcomes, loneliness, and perceptions of risk, to provide contextual information.

Ordinal survey items were grouped into binary variables for analysis. For example, AUDIT items were coded as positive if participants reported symptoms monthly or more often. Drinking behaviours were recoded using pre-specified thresholds (e.g., weekly or more frequent drinking). Demographic variables were treated as binary indicators (1 = Yes, 0 = No). Full item wordings and coding schemes are provided in Appendix A. For drinking prevalence, alcohol units were measured on a “typical” drinking day, with binge drinking characterised as 6 units in a single occurrence (women), or 8 (men) [30].

### 2.4. Statistical Analysis

All analyses applied survey weights provided by YouGov to align the sample with the UK population by age, sex, region, and social grade [29]. Despite 2464 respondents having met the inclusion criteria, weighting resulted in a slightly larger effective sample (*n* = 2486), which is used in the results. Cluster sizes therefore reflect weighted counts. Primary analyses focused on differences in drinking motives (captured in the dataset under variable A11) and drinking occasions (variable A5_new) between clusters and the benchmark of all increasing- and high-risk drinkers. Secondary analyses explored demographics (age, sex, social grade, IMD decile), drinking behaviours (frequency, typical units, binge drinking), and contextual indicators (symptoms of dependence, adverse outcomes, loneliness, social pressures, and perceived risk). Differences between cluster and benchmark proportions were analysed using two-proportion z-tests, with Benjamini–Hochberg false discovery rate [31] (FDR) correction applied for multiple comparisons; significance threshold was set at adjusted *p* < 0.05. All analyses were conducted in R (version 2025.05.1+513).

Survey data were cleaned prior to analysis: categorical responses were re-coded in line with the Drinkaware Monitor questionnaire. Results are reported as weighted percentages, percentage-point differences, and adjusted *p*-values.

### 2.5. Ethical Considerations

The Drinkaware Monitor 2023 dataset was anonymised prior to provision for secondary analysis and contained no personally identifiable information. The survey was conducted by YouGov in accordance with the Market Research Society Code of Conduct [32]. As analysis was undertaken on anonymised secondary data, no further ethical approval was required.

### 2.6. Acknowledgements

The authors acknowledge Drinkaware for providing access to the weighted Drinkaware Monitor 2023 dataset, including pre-segmented cluster outputs. Drinkaware receives voluntary, unrestricted donations from UK alcohol producers, pub operators, restaurants, supermarkets, and other retailers [33].

## 3. Results

A weighted sample of *n* = 2486 increasing- and high-risk drinkers was analysed; cluster sizes ranged from *n* = 140 (Cluster 5) to *n* = 817 (Cluster 2). Demographic characteristics are presented in Table 1. The benchmark compared against is the total group of all increasing- and high-risk drinkers. Drinking motivations and occasions are shown in Table 2 and Table 3, respectively, with weekly drinking prevalence in Figure 1. Contextual indicators (symptoms of dependence, adverse outcomes, and loneliness) are shown in Figure 2. To avoid overcrowding the main table, 95% confidence intervals, and adjusted *p*-values are provided in Appendix A.

Older respondents were overrepresented in Cluster 1 (aged 55+: 33.7% vs. 26.7%, adjusted *p* = 0.006; Table 1). Their motives emphasised product and taste (e.g., “because there are certain products you particularly enjoy”: 81.9% vs. 60.5%, adjusted *p* < 0.001) while social facilitation motives such as “because it makes you more outgoing” were lower (9.7% vs. 33.5%, adjusted *p* < 0.001; Table 2). They were more likely to drink in almost all occasion contexts apart from “home alone” or during “going out for a couple of drinks in the afternoon” (e.g., “several drinks at home with people in my household”: 65.8% vs. 52.4%, adjusted *p* < 0.001; Table 3). Weekly drinking prevalence of 4 or more times was also higher than the benchmark (42.8% vs. 34.0%, adjusted *p* = < 0.001; Figure 1). Cluster 1 were less likely to report feelings of “guilt or remorse after drinking” (12.8% vs. 18.6%, adjusted *p* = 0.003; Appendix A) or “feel that you lack companionship” (37.4% vs. 52.3%, adjusted *p* < 0.001; Figure 2).

Cluster 2 showed no significant demographic differences versus the benchmark (Table 1). Their drinking motives were significantly lower across every variable, except “to help you unwind,”, “because you like the taste”, “because it is satisfying to have a high-quality drink,” “because it pairs well with food,” and “because there are certain products you particularly enjoy” which did not reach significance (Table 2). They were more likely to drink “at home alone” (54.4% vs. 47.3%, adjusted *p* = 0.001) and when “going out for a meal” (51.2% vs. 46.3%, adjusted *p* = 0.021; Table 3). Weekly drinking prevalence of 4 or more occasions was lower (60.6% vs. 67.3%, adjusted *p* = 0.003; Figure 1), and fewer reported injury related to drinking (2.5% vs. 4.5%, adjusted *p* = 0.031; Appendix A).

Cluster 3 was overrepresented by younger participants (18–34: 46.8% vs. 35.6%, adjusted *p* < 0.001), and was more ethnically diverse (ethnic minority: 11.4% vs. 6.8%, adjusted *p* = 0.012). They displayed markedly higher social and identity motives such as “to put you at ease with people” (94.0% vs. 36.2%, adjusted *p* < 0.001; Table 2) and “because you feel more self-confident and sure of yourself” (92.0% vs. 35.5%, adjusted *p* < 0.001). Although weekly drinking prevalence of 4 or more occasions was lower (27.9% vs. 34.0%, adjusted *p* = 0.049; Figure 1), they showed higher likelihood of drinking in almost all occasion contexts, except “at home alone” where prevalence did not differ from the benchmark (Table 3). Additionally, feelings of loneliness were higher in Cluster 3, such as “feel that you lack companionship” (60.3% vs. 52.3%, adjusted *p* = 0.020; Figure 2).

Together, Clusters 1–3 highlight contrasting patterns: Cluster 1 defined by product and taste motives, Cluster 2 by lower social motives and more solitary drinking, and Cluster 3 by strong social and identity drivers, with wider drinking occasion coverage and higher loneliness.

Cluster 4 skewed younger (18–34: 53.3% vs. 35.6%, adjusted *p* < 0.001; Table 1), with more non-parents (66.8% vs. 51.1%, adjusted *p* < 0.001). They reported high social and identity motives (e.g., “because you feel more self-confident and sure of yourself”: 87.1% vs. 35.5%, adjusted *p* < 0.001; Table 2) but lower product and taste motives (e.g., “because it pairs well with food”: 7.7% vs. 35.4%, adjusted *p* < 0.001). They were less likely to drink “at home alone” (30.0% vs. 47.3%, adjusted *p* < 0.001) or “several drinks at home with people in my household” (35.7% vs. 52.4%, adjusted *p* < 0.001; Table 3). Adverse outcomes were more common (Appendix A), with higher reports of drinking-related injuries (11.2% vs. 4.5%, adjusted *p* < 0.001) and feeling “isolated from others” (71.2% vs. 53.9%, adjusted *p* < 0.001), with lower perceived alcohol-related health risk (32.0% vs. 45.5%, adjusted *p* < 0.001). Responses within Cluster 4 were less consistent than in other clusters, with indicators spanning both high social-motivation items and higher levels of emotionally driven outcomes (Appendix A).

Cluster 5 was overrepresented by older participants (aged 55+: 42.7% vs. 26.7%, adjusted *p* < 0.001) of those of C2DE social grade (56.3% vs. 42.0%, adjusted *p* < 0.001; Table 1). Their motives were lower across most variables, including social and dining-related motives (e.g., “to celebrate a special occasion with friends”: 20.3% vs. 65.0%, adjusted *p* < 0.001; “because it pairs well with food”: 20.2% vs. 35.4%, adjusted *p* < 0.001), but higher for coping (e.g., “to help you unwind”: 76.4% vs. 51.4%, adjusted *p* < 0.001; Table 2). Weekly drinking prevalence of 4 or more times was higher (46.8% vs. 34.0%, adjusted *p* = 0.006; Figure 1) and they were more likely to drink “at home alone” (69.9% vs. 47.3%, adjusted *p* < 0.001) but reported lower prevalence across all other occasions (Table 3). They showed higher perceived alcohol-related health risk (59.6% vs. 45.5%, adjusted *p* = 0.010) and reported less peer pressure (“being asked to explain or justify why you aren’t drinking alcohol”: 14.4% vs. 23.5%, adjusted *p* = 0.038; Appendix A).

Similarly, Cluster 6 also presented older (aged 55+: 37.4% vs. 26.7%, adjusted *p* = 0.004) and more likely to be social grade C2DE (53.6% vs. 42.0%, adjusted *p* = 0.005; Table 1). Their motivations were lower across all variables including taste, social, dining-related and coping domains (e.g., “because you like the taste” (34.6% vs. 76.0%, adjusted *p* < 0.001; “to celebrate a special occasion with friends” (29.7% vs. 65.0%, adjusted *p* < 0.001; Table 2). Drinking was also less common across every measured occasion, including at home (“several drinks at home with people in my household”: 36.5% vs. 52.4%, adjusted *p* < 0.001) and at events (22.0% vs. 37.7%, adjusted *p* < 0.001; Table 3). Adverse consequences included greater reports of an alcohol-related injury (8.5% vs. 4.5%, adjusted *p* = 0.021), though they were less likely to report “a feeling of guilt or remorse after drinking” (12.5% vs. 18.6%, adjusted *p* = 0.043; Appendix A).

Clusters 4–6 reinforce that while demographic differences are evident, the strongest variation arises in motives and drinking occasions, which underpin differences in symptoms of dependence and related outcomes.

## 4. Discussion

This study segmented increasing- and high-risk drinkers in the UK into six clusters defined primarily by drinking motives and drinking occasions, with demographic, behavioural, and psychosocial indicators examined to provide context. The findings confirmed that substantial heterogeneity exists within this population, with clusters differing in the reasons they drink and the contexts in which drinking occurs. These motivational and situational differences underpin variation in symptoms of dependency, loneliness, and risk perception, emphasising that harmful drinking is shaped by diverse drivers. Universal strategies remain important, but tailored interventions that account for motive- and context-specific pathways may be more effective.

### 4.1. Interpretation and Implications

A key contribution of this study is its focus on increasing- and high-risk drinkers, a group often neglected in both research and practice. Public health interventions in the UK typically target either dependent drinkers, through specialist treatment services, or the whole population, through measures such as pricing, availability, and awareness campaigns [14,15,17]. As a result, the large group of drinkers who fall into the increasing- and high-risk categories often receive little tailored attention, despite representing almost a quarter of the population and experiencing substantial alcohol-related harm [11]. The prevention paradox [20] illustrates why this group is so important: although dependent drinkers face the greatest individual danger, most alcohol-related harm arises from the much larger population of increasing- and high-risk drinkers. Recognising the diversity across this group may therefore help inform more responsive and effective policy approaches.

The results highlight the extent of this heterogeneity. Clusters 1 and 3 both consumed alcohol regularly and were significantly more likely than the benchmark to drink across a broad range of occasions. However, the drivers and risks underpinning this behaviour differed greatly. Cluster 1, an older and predominantly white subgroup, was motivated by product and taste, drank more frequently on a weekly basis and often at home with household members, reporting fewer psychosocial vulnerabilities. By contrast, Cluster 3, a younger and more diverse group, was strongly socially motivated, drank more often in out-of-home settings such as events and nights out, and showed higher loneliness, peer pressure, and symptoms of dependence. Whereas Cluster 1’s consumption reflected routine and embedded home-based drinking, Cluster 3’s profile was characterised by socially driven and emotionally vulnerable patterns. This comparison illustrates how groups with similarly broad drinking occasions can nevertheless present very different psychosocial profiles, requiring equally different interventions. Policies that influence product value and marketing (e.g., MUP, labelling, advertising restrictions) may be more relevant to Cluster 1 [34,35,36,37], while younger socially motivated groups like Cluster 3 may benefit from community-based programmes, alcohol-free leisure alternatives, and digital tools that provide non-alcoholic means of social connection [38]. Although both Clusters 3 and 4 were socially motivated, Cluster 3 was more strongly linked with loneliness and peer pressure, whereas Cluster 4 was marked by lower risk perception and higher injury, suggesting different intervention priorities. These patterns also show that motivational and contextual factors can overlap across segments. For example, while Cluster 1 and Cluster 3 differ clearly in age and social context, both display broad drinking occasions arising from distinct underlying drivers. This reinforces that the clusters are heuristic rather than mutually exclusive categories.

Differences were also apparent between Clusters 2 and 5. Both groups preferred to drink alone at home, but their motives and contexts diverged. Cluster 2 showed no demographic differences from the benchmark and reported lower social and enhancement motives, suggesting a risk of normalised but less harmful solitary consumption. Weekly drinking prevalence was not elevated, and adverse outcomes were relatively limited, implying that light-touch strategies such as awareness campaigns about home drinking [39] or unit measures may be appropriate, such as the ‘Know Your Units’ campaign in Northern Ireland [40]. By contrast, Cluster 5 was older and more likely to be of C2DE social grade, combined solitary home drinking with higher coping motives, and showed significantly higher weekly drinking prevalence. This profile points to alcohol use for stress or emotional regulation, indicating a need for more intensive responses such as mindfulness-based support, integrated mental health and alcohol services, or workplace wellbeing programmes [41,42,43]. Thus, even where drinking occasions overlap (e.g., drinking at home), demographic, motivational, and behavioural differences imply the need for tailored intervention strategies.

Clusters 4 and 6 shared relatively low endorsement of product- and taste-related motives and fewer routine drinking occasions (e.g., “several drinks at home with people in my household”), yet both reported higher adverse consequences, underlining different pathways to harm. Cluster 4, a younger subgroup who were less likely to be parents, showed high social and identity motives but low taste and product motives alongside reduced risk perception and lower willingness to cut down. Their drinking occasions were less home-based and more often in out-of-home contexts, but adverse outcomes such as alcohol-related injury and loneliness were more common. This group may benefit from psychosocial interventions such as motivational interviewing, but with messaging that directly addresses the social normalisation of heavy drinking and provides alternative forms of social connection [44]. By contrast, Cluster 6 was older and more likely to be of C2DE social grade, characterised by consistently lower motives across all domains and fewer drinking occasions across contexts. Although adverse outcomes were mixed, with higher reports of alcohol-related injuries but less feelings of “guilt or remorse”, their consistently low endorsement of drinking motives suggests this group may be less responsive to interventions targeting social or coping drivers. Alternative approaches, such as clearer risk communication, proactive clinical screening, and digitally delivered brief interventions, may therefore be more effective [38]. Even within older adults, profiles diverged: Cluster 1’s taste-driven drinking, Cluster 5’s coping-motivated drinking, and Cluster 6’s low-motive profile each imply different intervention needs, underlining that there are distinct differences among older drinkers.

Together, these comparisons highlight that increasing- and high-risk drinkers cannot be treated as a homogeneous group. Even among individuals who drink in the same settings, underlying motives and psychosocial profiles differ, meaning that tailored strategies are essential if interventions are to achieve meaningful impact.

Although both biological sex and gender identity were collected, gender-stratified analyses were not conducted due to the exploratory scope of this segmentation. Given well-established gender differences in drinking motives, future work could examine whether the latent classes or their underlying motivations differ by gender. The survey also included validated wellbeing measures (loneliness, depressive affect, anxiety), but these were excluded to keep the segmentation focused on core motivational and behavioural indicators. Similarly, variables such as financial stress and attitudes toward alcohol policy were omitted to avoid broadening the model beyond its intended scope. Incorporating these contextual and psychosocial factors in future analyses could offer a more integrated understanding of the pressures shaping drinking behaviour.

### 4.2. International Approaches and Relevance

The multifaceted nature of these clusters highlights the value of tailoring interventions to subgroup characteristics. International evidence offers examples of how motive- and context-sensitive approaches can complement universal policies. Personality-targeted programmes such as PreVenture demonstrate that addressing psychosocial vulnerabilities can reduce binge drinking in adolescents [45,46], suggesting potential for adaptation to adult subgroups defined by coping or social motives. The Icelandic Prevention Model underscores the effectiveness of strengthening community and social infrastructure, though its reliance on high levels of coordination limits transferability [47,48]. Digital tailoring, including apps, may be useful for younger, socially motivated drinkers [49], but are less suitable for older or more deprived groups.

While the present study does not assess intervention effectiveness, the cluster characteristics offer a basis for generating hypotheses about which subgroups may benefit from particular forms of support. For instance, groups with stronger coping-related motives may require approaches focused on emotional regulation and wellbeing, whereas socially driven groups might respond better to peer-oriented messaging or alcohol-free social alternatives. These insights may be relevant for primary care, community programmes, and public health campaigns that aim to move beyond broad, one-size-fits-all messaging.

Implementing segmented approaches in practice presents challenges. Tailored strategies may require greater up-front resources and careful attention to identifying target groups accurately. However, if such approaches improve acceptability and engagement among specific subgroups, the additional investment may be justified. The present segmentation therefore provides a starting point for future work exploring how motive- and occasion-based tailoring could complement existing UK alcohol policies.

### 4.3. Strengths and Limitations

A key strength of this study is the focus on increasing- and high-risk drinkers, a group typically overlooked in favour of dependent drinkers or the general population. By using motives and occasions as the primary basis for segmentation, this study extends previous work that relied on adolescent-oriented tools [26] or location-only approaches [50] and adds to prior segmentation research on UK drinkers [28]. To the best of our understanding, this study is the first in the UK to apply an adult-validated motives tool (DMQ-A) combined with drinking occasion data to cluster increasing- and high-risk drinkers, providing novel insight into an overlooked group. Examining symptoms of dependency, loneliness, and related outcomes as contextual factors adds nuance without confounding the underlying segmentation.

Several limitations should be acknowledged. The cross-sectional design prevents causal inference, meaning it is not possible to establish whether factors such as loneliness are causes or consequences of harmful drinking, nor to assess how individuals may transition between clusters over time. Alcohol consumption was self-reported, raising the possibility of recall and social desirability biases. Such surveys typically capture only 40–60% of actual consumption when compared with sales data, indicating a high likelihood of under-reporting in the present study [51,52,53]. Online panel recruitment may also under-represent groups with limited digital access. This study uses secondary data, and therefore the selection and measurement of variables were constrained by the existing survey design. Several potentially influential variables such as detailed mental health indicators, income, and cultural background, were not included in this initial exploratory cluster analysis. Similarly, gender-stratified motive analyses and wellbeing-based analyses were not conducted, and this may limit the interpretability of certain clusters. There was notable heterogeneity within some clusters, particularly Cluster 4, which suggests that further sub-structure may exist.

### 4.4. Future Directions

Future research should investigate whether individuals transition between motive-driven clusters over time. Such transitions may be tied to life-course shifts in roles and responsibilities: prior work demonstrates that alcohol use changes with age and role transitions (e.g., employment, marriage, parenthood) [54], and large longitudinal cohorts show drinking trajectories that peak in early adulthood before declining [55,56]. Evaluating interventions tailored to specific motives and occasions (e.g., stress management for coping-driven drinkers, social alternatives for identity-driven groups) against generic approaches will be essential. Linking segmentation to broader policy levers, such as pricing and availability restrictions, would also clarify how tailored and population-level strategies interact. Such evidence will help translate segmentation into practical tools for alcohol harm reduction.

## 5. Conclusions

This study identified six clusters of increasing- and high-risk drinkers, defined by drinking motives and occasions and contextualised by demographics and psychosocial outcomes. These ranged from taste-motivated drinkers (Cluster 1), to solitary home drinkers (Cluster 2), socially driven groups (Clusters 3 and 4), and older subgroups with coping or low-motive patterns (Clusters 5 and 6). Comparisons between clusters further demonstrate that even drinkers who consume alcohol in similar contexts (e.g., at home) may differ substantially in their underlying motivations and vulnerabilities, underscoring the heterogeneity of this population.

The segmentation presented here should be interpreted as exploratory and hypothesis-generating rather than as evidence for intervention effectiveness. Nonetheless, the findings highlight potential subgroups that may merit further investigation and point to ways in which universal alcohol policies could be complemented by more tailored approaches aligned with motive- and context-specific pathways.

Importantly, this study emphasises the value of focusing on increasing- and high-risk drinkers, a large but often overlooked population who contribute disproportionately to alcohol-related harm in line with the prevention paradox. Understanding their diversity represents an important step towards developing more nuanced strategies for reducing alcohol-related harm in the UK.

## Figures and Tables

**Figure 1 nutrients-17-03745-f001:**
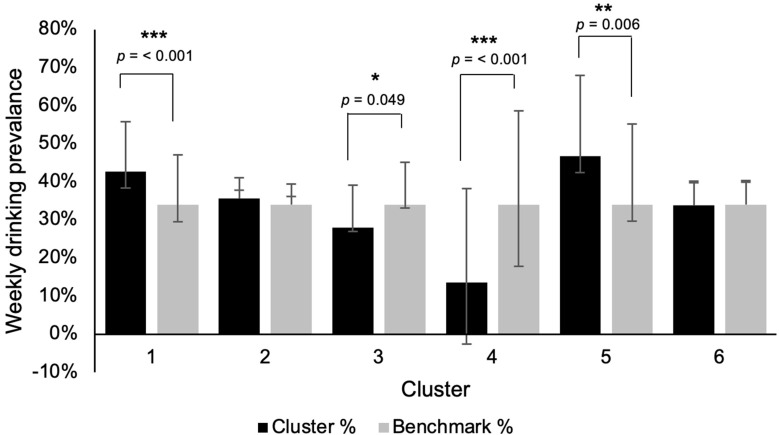
Weekly drinking prevalence (A1; positive = ≥4 times per week) by cluster against benchmark (all increasing- and high-risk drinkers) as weighted proportions with 95% confidence intervals. * adjusted *p* < 0.05, ** *p* < 0.01, *** *p* < 0.001.

**Figure 2 nutrients-17-03745-f002:**
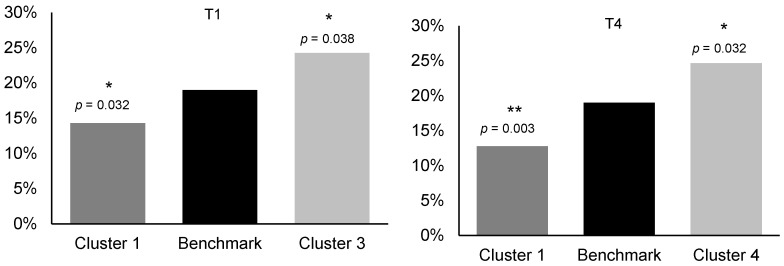
Symptoms of dependence (T1; positive = monthly or more), adverse drinking consequences (T4, T6; positive = yes in the past year), and loneliness (MH_5_1; positive = occasionally or more often) significantly higher or lower than the benchmark (all increasing- and high-risk drinkers—indicated by middle bar). * adjusted *p* < 0.05, ** *p* < 0.01, *** *p* < 0.001. Full statistics are provided in Appendix A.

**Table 1 nutrients-17-03745-t001:** Demographics by cluster as weighted proportions (%) against benchmark (all increasing- and high-risk drinkers). * adjusted *p* < 0.05, ** *p* < 0.01, *** *p* < 0.001.

Variable		Benchmark	Cluster 1	Cluster 2	Cluster 3	Cluster 4	Cluster 5	Cluster 6
		(*n* = 2486)	(*n* = 622)	(*n* = 817)	(*n* = 344)	(*n* = 315)	(*n* = 140)	(*n* = 248)
Gender	Male	61.36	58.2	65.54	62.49	52.19	65.82	63.09
	Female	38.64	41.8	34.46	37.51	47.81	34.18	36.91
Age	18–34	35.55	27.67 **	37.13	46.82 ***	53.29 ***	11.79 ***	25.45 ***
	35–54	37.71	38.66	36.34	39.36	34.59	45.56	37.14
	55+	26.73	33.66 **	26.54	13.82 ***	12.12 ***	42.65 ***	37.41 **
Ethnicity	Ethnic minority	6.77	3.86 *	7.25	11.41 *	8.53	3.32	5.72
	White	90.39	94.52 **	89.66	84.60 **	87.58	94.25	91.87
Parental status	Parent (U18)	23.71	24.67	23.31	28.55	19.71	25.77	19.83
	Parent (O18)	25.57	31.95 **	26.22	12.25 ***	15.6 **	37.11 *	32.09
	Non-parent	51.08	46.3	49.45	58.48	66.83 ***	36.19 **	46.64
Social grade	ABC1	57.97	63.34	60.5	58.28	55.92	43.75 **	46.4 **
	C2DE	42.03	36.66	39.5	41.72	44.08	56.25 **	53.6 **
IMD decile	1 to 3	30.53	26.28	32.22	31.7	28.96	28.96	36.58
	4 to 7	38.98	40.58	37.35	40.47	40.98	40.98	34.01
	8 to 10	30.49	33.14	30.43	27.83	30.06	30.06	29.41

**Table 2 nutrients-17-03745-t002:** Drinking motivations (A11; positive = “half of the time” or more) by cluster as weighted proportions (%) against benchmark (all increasing- and high-risk drinkers). * adjusted *p* < 0.05, ** *p* < 0.01, *** *p* < 0.001.

Response	Benchmark	Cluster 1	Cluster 2	Cluster 3	Cluster 4	Cluster 5	Cluster 6
Because drinking is part of the fun with family or friends	56.4	76.68 ***	39.31 ***	89.91 ***	67.47 ***	10.74 ***	27.15 ***
Because drinking adds a certain warmth to social occasions	56.85	78.59 ***	40.13 ***	90.81 ***	68.06 ***	16.16 ***	19.13 ***
To celebrate a special occasion with friends	64.97	83.76 ***	53.27 ***	90.81 ***	77.69 ***	20.27 ***	29.69 ***
To calm down when you are tense	30.01	24.7 *	24.81 **	59.32 ***	30.51	48.59 ***	8.76 ***
To help you unwind	51.42	52.31	48.72	83.39 ***	37.73 ***	76.44 ***	17.15 ***
To make you more outgoing	33.51	9.66 ***	23.87 ***	86.06 ***	80.63 ***	9.98 ***	5.91 ***
To overcome shyness	30.61	4.79 ***	19.94 ***	86.03 ***	79.02 ***	8.94 ***	4.58 ***
Because you feel more self-confident and sure of yourself	35.46	9.2 ***	24.48 ***	92.02 ***	87.09 ***	16.3 ***	4.54 ***
To put you at ease with people	36.22	11.62 ***	27.1 ***	93.96 ***	79.73 ***	13.43 ***	5.74 ***
Because it is satisfying to have a high-quality drink	52.17	64.84 ***	51.31	84.93 ***	31.7 ***	32.16 ***	15.22 ***
Because it pairs well with food	35.39	50.56 ***	37.49	51.59 ***	7.69 ***	20.24 ***	11.77 ***
Because there are certain products you particularly enjoy	60.47	81.91 ***	56.80	85.8 ***	35.22 ***	52.31	20.34 ***
Because you like the taste	76.01	92.78 ***	76.11	91.31 ***	58.08 ***	77.10	34.6 ***
Because it makes you happy	60.28	79.36 ***	43.44 ***	92.1 ***	67.06 *	69.79 *	9.86 ***
Because it gives you a pleasant feeling	66.66	84.64 ***	53.75 ***	93.77 ***	69.89	79.63 **	15.2 ***

**Table 3 nutrients-17-03745-t003:** Drinking occasions (A5_new; positive = “once a week or more” or “1–3 times a month”) by cluster as weighted proportions (%) against benchmark (all increasing- and high-risk drinkers). * adjusted *p* < 0.05, ** *p* < 0.01, *** *p* < 0.001.

Response	Benchmark	Cluster 1	Cluster 2	Cluster 3	Cluster 4	Cluster 5	Cluster 6
Drinking at home alone	47.29	42.75	54.35 ***	51.95	30.02 ***	69.87 ***	38.17 **
A small number of drinks at home with people in my household	56.53	70.14 ***	59.33	67.88 ***	37.17 ***	28.87 ***	37.62 ***
Several drinks at home with people in my household	52.36	65.77 ***	53.40	61.53 **	35.68 ***	29.79 ***	36.47 ***
Getting together at your or someone else’s house	37.26	42.86 *	37.84	47.03 ***	34.92	11.58 ***	25.27 ***
Going out for a meal	46.25	57.59 ***	51.2 *	52.11	30.5 ***	9.41 ***	34.23 ***
Evening or night out with friends	51.28	57.23 *	52.21	59.58 **	51.77	17.01 ***	40.54 **
Going out for a couple of drinks in the afternoon	30.79	27.82	34.71	43.38 ***	25.83	5.86 ***	28.28
Drinking at events	37.66	43.2 *	40.25	49.28 ***	33.16	7.22 ***	22.02 ***

## Data Availability

The raw data supporting the conclusions of this article will be made available by the authors on request.

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
