# Peer review of "Segmenting Increasing- and High-Risk Alcohol Drinkers by Motives and Occasions: Implications for Targeted Interventions"

_nutrients, 2025, doi:10.3390/nu17233745_

Round 1

Reviewer 1 Report

Comments and Suggestions for Authors

Thank you for the opportunity to review this manuscript. In line 42 ....rose by 8.8% between 2019 and 2022", could be because of the pandemy? I suggest the authors to specify what ABC1 and C2DE mean.

Author Response

Thank you very much for taking the time to review this manuscript. Please find the detailed responses below and the corresponding revisions highlighted in the re-submitted file.

Comments 1: Thank you for the opportunity to review this manuscript. In line 42 ....rose by 8.8% between 2019 and 2022", could be because of the pandemy? I suggest the authors to specify what ABC1 and C2DE mean.

Response 1: Thank you for this helpful comment. We have clarified in the Introduction that the rise in increasing- and high-risk drinking between 2019 and 2022 coincided with the COVID-19 pandemic, which is likely to have contributed to the observed increase. We have also added definitions of ABC1 and C2DE, specifying that these refer to standard UK social grade classifications based on occupation.

Reviewer 2 Report

Comments and Suggestions for Authors

This study aimed to define and characterise clusters of increasing-  and high-risk drinkers, based on drinking motives and occasions and contextualised by demographics and psychosocial factors, to inform tailored harm-reduction strategies. Segmentation was based on drinking occasions and drinking motives using the Drinking Motives Questionnaire for Adults (DMQ-A).  And the main idea is that segmentation can support the development of more tailored and effective strategies. The study discovered six distinct subgroups. It was concluded that increasing- and high-risk drinkers are heterogeneous, shaped by different motives and contexts. Tailoring interventions to subgroup profiles, such as socially motivated younger adults and older adults whose drinking is driven by coping motives, may strengthen alcohol harm-reduction strategies.

I find this topic original and highly relevant to the field. This article addresses a specific gap in the field.  Namely, earlier analyses of Drinkaware Monitor data grouped drinkers by drinking motivations and occasions but relied on the Drinking Motives Questionnaire - Revised Short Form (DMQ-R SF), a tool developed for an adolescent population, to achieve this. This reliance may have overlooked important drivers of adult drinking. The present study is the first to address this gap by combining drinking occasion data with the Drinking Motives Questionnaire for Adults (DMQ-A), an adult-adapted motives questionnaire, in a novel approach to identify six distinct clusters of increasing- and high-risk drinkers. It was hypothesised that clusters would show clear and systematic differences, indicating that interventions tailored to motives and drinking contexts could be more effective than generic approaches.

A key contribution of this study is its focus on increasing- and high-risk drinkers, a group often neglected in both research and practice. Public health interventions in the UK typically target either dependent drinkers, through specialist treatment services, or the whole population, through measures such as pricing, availability, and awareness campaigns. As a result, the large group of drinkers who fall into the increasing- and high-risk categories often receive little tailored attention, despite representing almost a quarter of the population and experiencing substantial alcohol-related harm. The prevention paradox illustrates why this group is so important: although dependent drinkers face the greatest individual danger, most alcohol-related harm arises from the much larger population of increasing- and high-risk drinkers. Recognising the diversity across this group may therefore help inform more responsive and effective policy approaches.

The methodology is very well described, and the authors have sincerely elaborated on the limitations of their research.

The Conclusions are consistent with the evidence and arguments presented, and they address the main question posed.

The references are appropriate, and the reference list is sufficiently extensive.

Author Response

Thank you very much for taking the time to review this manuscript. Please find the detailed responses below and the corresponding revisions highlighted in the re-submitted file.

Comments 1: This study aimed to define and characterise clusters of increasing-  and high-risk drinkers, based on drinking motives and occasions and contextualised by demographics and psychosocial factors, to inform tailored harm-reduction strategies. Segmentation was based on drinking occasions and drinking motives using the Drinking Motives Questionnaire for Adults (DMQ-A).  And the main idea is that segmentation can support the development of more tailored and effective strategies. The study discovered six distinct subgroups. It was concluded that increasing- and high-risk drinkers are heterogeneous, shaped by different motives and contexts. Tailoring interventions to subgroup profiles, such as socially motivated younger adults and older adults whose drinking is driven by coping motives, may strengthen alcohol harm-reduction strategies.

I find this topic original and highly relevant to the field. This article addresses a specific gap in the field.  Namely, earlier analyses of Drinkaware Monitor data grouped drinkers by drinking motivations and occasions but relied on the Drinking Motives Questionnaire - Revised Short Form (DMQ-R SF), a tool developed for an adolescent population, to achieve this. This reliance may have overlooked important drivers of adult drinking. The present study is the first to address this gap by combining drinking occasion data with the Drinking Motives Questionnaire for Adults (DMQ-A), an adult-adapted motives questionnaire, in a novel approach to identify six distinct clusters of increasing- and high-risk drinkers. It was hypothesised that clusters would show clear and systematic differences, indicating that interventions tailored to motives and drinking contexts could be more effective than generic approaches.

A key contribution of this study is its focus on increasing- and high-risk drinkers, a group often neglected in both research and practice. Public health interventions in the UK typically target either dependent drinkers, through specialist treatment services, or the whole population, through measures such as pricing, availability, and awareness campaigns. As a result, the large group of drinkers who fall into the increasing- and high-risk categories often receive little tailored attention, despite representing almost a quarter of the population and experiencing substantial alcohol-related harm. The prevention paradox illustrates why this group is so important: although dependent drinkers face the greatest individual danger, most alcohol-related harm arises from the much larger population of increasing- and high-risk drinkers. Recognising the diversity across this group may therefore help inform more responsive and effective policy approaches.

The methodology is very well described, and the authors have sincerely elaborated on the limitations of their research.

The Conclusions are consistent with the evidence and arguments presented, and they address the main question posed.

The references are appropriate, and the reference list is sufficiently extensive.

Response 1: We sincerely appreciate the reviewer’s positive assessment of the manuscript and are grateful for the thoughtful summary of its contribution. No changes were required in response to this comment.

Reviewer 3 Report

Comments and Suggestions for Authors
  1. The supplementary materials confirm that both biological sex and gender identity were collected, and Table 1 in the main manuscript reports male/female proportions for each cluster. However, these variables were not included in the motive analysis (Table 2), where all responses are presented in aggregate. Since gender differences in enhancement and coping motives are well established, the lack of stratified or interaction analysis represents a missed analytical opportunity rather than a data limitation. At minimum, the authors should discuss potential gender effects or justify why these were not examined.
  2. The survey included validated wellbeing measures that capture loneliness, depressive affect, and anxiety—factors directly linked to drinking motives. Their omission is particularly notable for Cluster 4, which shows both high social motives and elevated adverse outcomes. Incorporating these indicators or acknowledging their absence would strengthen the interpretation.
  3. Both Figure 1 and Supplementary Table S2 show considerable within-cluster variability for Cluster 4. The group appears to mix socially motivated and emotionally compensatory drinking patterns. A brief acknowledgement of this potential sub-structure would improve interpretive transparency and prevent overgeneralization.
  4. Provide effect sizes or confidence intervals for key differences instead of relying solely on p-values.
  5. The survey contained multiple items on financial stress and policy attitudes. These could have been linked to drinking patterns but were entirely omitted from the published results. Including these would enrich interpretation, especially given ongoing UK policy debates on affordability and harm reduction.
  6. The conclusions somewhat overstate the applied implications of the results. While the segmentation highlights potential subgroups for targeted interventions, the study does not empirically test intervention responsiveness or behavioral change. The conclusions should be reframed to reflect hypothesis generation rather than demonstrated impact.

Author Response

Thank you very much for taking the time to review this manuscript. Please find the detailed responses below and the corresponding revisions highlighted in the re-submitted file.

Comments 1: The supplementary materials confirm that both biological sex and gender identity were collected, and Table 1 in the main manuscript reports male/female proportions for each cluster. However, these variables were not included in the motive analysis (Table 2), where all responses are presented in aggregate. Since gender differences in enhancement and coping motives are well established, the lack of stratified or interaction analysis represents a missed analytical opportunity rather than a data limitation. At minimum, the authors should discuss potential gender effects or justify why these were not examined.

Response 1: Thank you for highlighting this important point. We agree that gender differences in enhancement and coping motives are well established. As suggested, we have now added explicit justification in the Discussion explaining that gender-stratified modelling was not conducted due to the exploratory scope of the segmentation approach and to avoid over-parameterisation of the latent class model. We have also added a statement acknowledging that gender-related differences are plausible and represent a valuable avenue for future analyses using this dataset.

Comments 2: The survey included validated wellbeing measures that capture loneliness, depressive affect, and anxiety—factors directly linked to drinking motives. Their omission is particularly notable for Cluster 4, which shows both high social motives and elevated adverse outcomes. Incorporating these indicators or acknowledging their absence would strengthen the interpretation.

Response 2: Thank you - we appreciate this observation. We have added a paragraph in the Discussion noting that these wellbeing indicators were available but intentionally omitted to maintain focus on core motivational and behavioural variables. We now explicitly acknowledge that their inclusion could enrich interpretation in future analysis - particularly for groups such as Cluster 4 - and outline how potential studies could examine how wellbeing profiles align with or differentiate the identified clusters.

Comments 3: Both Figure 1 and Supplementary Table S2 show considerable within-cluster variability for Cluster 4. The group appears to mix socially motivated and emotionally compensatory drinking patterns. A brief acknowledgement of this potential sub-structure would improve interpretive transparency and prevent overgeneralization.

Response 3: Thank you for this valuable point. We have now added text acknowledging that Cluster 4 displayed greater within-cluster variability compared to other classes, potentially reflecting a mix of socially motivated and emotionally compensatory drinking patterns. This clarification is included in the Interpretation subsection to increase transparency and avoid over-generalisation.

Comments 4: Provide effect sizes or confidence intervals for key differences instead of relying solely on p-values.

Response 4: Thank you for this important suggestion. The 95% confidence intervals were incorporated into Supplementary Table 2 and this has been highlighted early in the results section to support more informative interpretation.

Comments 5: The survey contained multiple items on financial stress and policy attitudes. These could have been linked to drinking patterns but were entirely omitted from the published results. Including these would enrich interpretation, especially given ongoing UK policy debates on affordability and harm reduction.

Response 5: We agree that these variables could have offered useful contextual insight. We have added a paragraph in the Discussion explaining that these variables were considered but excluded to keep the segmentation model focused on core motivational and behavioural indicators. We now explicitly note that financial stress and policy attitudes represent promising areas for future analyses, particularly given their relevance to UK alcohol affordability debates.

Comments 6: The conclusions somewhat overstate the applied implications of the results. While the segmentation highlights potential subgroups for targeted interventions, the study does not empirically test intervention responsiveness or behavioral change. The conclusions should be reframed to reflect hypothesis generation rather than demonstrated impact.

Response 6: Thank you for this helpful guidance. We have revised the Conclusion to emphasise that the segmentation provides a hypothesis-generating framework rather than direct evidence of intervention effectiveness. The revised conclusion now clearly states that the findings suggest possible avenues for targeted strategies, which require future testing.

Reviewer 4 Report

Comments and Suggestions for Authors

Manuscript ID: nutrients-3988391

Title: “Segmenting increasing-and high-risk alcohol drinkers by motives and occasions: implications for targeted interventions”

The following aspects are suggested for consideration:

  1. The authors should justify why they selected the final number of six groups.

  1. It is suggested that the limitations of using a secondary analysis of the Drinkaware Monitor 2023 survey be indicated.

  1. How were potentially influential variables not included, such as mental health, income, or cultural patterns, handled?

  1. Is there a risk of overlap between motivational and contextual clusters that complicates their applicability in intervention programmes?

  1. According to the authors, what characteristics of the clusters could help prioritise public health actions? How could the findings be applied in primary care services, community programmes, or specific campaigns?

  1. What challenges might arise when attempting to implement segmented interventions in practice?

Author Response

Thank you very much for taking the time to review this manuscript. Please find the detailed responses below and the corresponding revisions highlighted in the re-submitted file.

Comments 1: The authors should justify why they selected the final number of six groups.

Response 1: Thank you for raising this point. We have expanded the Methods section to clarify that the six-cluster model was selected based on clear interpretability and explicitly note that the final decision was guided by conceptual clarity.

Comments 2: It is suggested that the limitations of using a secondary analysis of the Drinkaware Monitor 2023 survey be indicated.

Response 2: We agree with this suggestion and have added text to the Limitations section noting that the study is constrained by the scope and structure of the existing Monitor questionnaire. We now explicitly acknowledge that variable selection was limited to what had been collected and that measurement timing and self-report bias are inherent constraints of secondary survey analysis.

Comments 3: How were potentially influential variables not included, such as mental health, income, or cultural patterns, handled?

Response 3: Thank you for this helpful question. We have clarified in the Methods and Discussion that these variables were considered but excluded to maintain clarity. Their omission is now acknowledged as a limitation, and we note that incorporating these broader contextual factors represents a meaningful direction for future analyses.

Comments 4: Is there a risk of overlap between motivational and contextual clusters that complicates their applicability in intervention programmes?

Response 4: We appreciate this insightful comment. We have added a sentence in the Discussion noting that some conceptual overlap is expected, as drinking motivations and contexts are inherently interrelated. We emphasise that the clusters should be interpreted as heuristic groupings rather than discrete categories and that some overlap does not preclude their usefulness for hypothesis generation.

Comments 5: According to the authors, what characteristics of the clusters could help prioritise public health actions? How could the findings be applied in primary care services, community programmes, or specific campaigns?

Response 5: Thank you for highlighting the need for clearer applied implications. We have expanded the Discussion to provide concrete examples: e.g., tailoring behavioural support to coping-motivated older adults, social-norm messaging for socially driven younger adults, and risk-communication strategies for low-motive older adults. We also note how primary care screening, community-based initiatives, and targeted campaigns could use cluster characteristics to prioritise outreach.

Comments 6: What challenges might arise when attempting to implement segmented interventions in practice?

Response 6: We agree that practical challenges should be acknowledged. We have now added a brief section outlining key barriers, including resource demands, the risk of misclassifying individuals, the need for scalable tools to identify subgroups, and ensuring acceptability among diverse populations. These points are added to provide a balanced view of the potential and limitations of segmented approaches.

Round 2

Reviewer 3 Report

Comments and Suggestions for Authors

the authors have now improved the paper, therefore, I suggest its publication. thank you